# Matrix-Based Method for Inferring Elements in Data Attributes Using a Vector Space Model

**Teruaki Hayashi \*** and **Yukio Ohsawa**

Department of Systems Innovation, School of Engineering, The University of Tokyo, Tokyo 113-8656, Japan; ohsawa@sys.t.u-tokyo.ac.jp
\* Correspondence: hayashi@sys.t.u-tokyo.ac.jp

**Abstract:** This article addresses the task of inferring elements in the attributes of data. Extracting data related to our interests is a challenging task. Although data on the web can be accessed through free text queries, it is difficult to obtain results that accurately correspond to user intentions because users might not express their objects of interest using exact terms (variables, outlines of data, etc.) found in the data. In other words, users do not always have sufficient knowledge of the data to formulate an effective query. Hence, we propose a method that enables the type, format, and variable elements to be inferred as attributes of data when a natural language summary of the data is provided as a free text query. To evaluate the proposed method, we used the Data Jacket's datasets whose metadata is written in natural language. The experimental results indicate that our method outperforms those obtained from string matching and word embedding. Applications based on this study can support users who wish to retrieve or acquire new data.

**Keywords:** data jacket; variable label; metadata; natural language processing; market of data; vector space model

---

## 1. Introduction

The global trends of big data and artificial intelligence (AI) have introduced various types of data that cannot be handled by the existing analytical technologies; thus, attention on areas not centered on AI technologies has increased. Rather than relying on a single data source, methods have been proposed to solve such problems and obtain new values in data through the distribution, exchange, and linking of the data across various fields. With improvements in catalogs and portal sites available in the data markets, opportunities for users to obtain data from data holders and providers have increased. Therefore, a data market has been developed in which various stakeholders exchange data and information about the data across different fields [1,2]. In particular, the developments of the Internet of things and cloud computing, and the privilege of mobile, digital markets for data have emerged [3,4]. Various stakeholders have discussed the potential benefits of reusing and analyzing massive amounts of data [5,6]. However, these typically affect data privacy and security [7–10]. Moreover, it is often difficult to obtain and utilize data that are specifically related to our interests. Even if relevant information is available publicly on the web, users may find it challenging to specify areas of interest owing to information overload. From the perspective of limited human cognition, it has been observed that excessive information renders it difficult for human decision makers to derive the necessary information and discover useful knowledge [11]. Therefore, a support system is required to obtain information related to user interests.

Another related issue is the difficulty encountered by users in obtaining data that accurately correspond to their intentions because users might not express their objects of interest using the exact terms (names of variables, outlines, etc.) used in the relevant data [12]. A user wishing to obtain

new data for a business focusing on foreign tourists, for example, may obtain street interviews and questionnaires completed by foreigners. However, if specific information of interest, specifically the nationality of foreigners, is missing from the acquired data, the procedure may have to be repeated. Owing to the costs involved in obtaining data, a reworking procedure should be avoided after data acquisition. In product management, reworking in the latter stage of a product design has been recognized as a serious risk [13]. To avoid this risk in data design and management, it is important to specify the exact data that should be obtained to implement effective decision making.

Hayashi and Ohsawa [14,15] proposed a method called a Variable Quest (VQ) for inferring variables from a data outline when information on the variables are missing or unknown. Here, variables refer to one of the attributes of the data; for example, "latitude" or "longitude" are elements in the variables. The proposed method infers variables that may be present in the data by inputting the summary of the data presented in natural language. Information on the variables and data outline is extracted from the Data Jacket's dataset. A data jacket (DJ) is a technique for sharing information on data without exposing the data itself by describing a summary of the data in natural language [16]. The idea of a DJ is to share a "summary of the data" as metadata while reducing the data management cost and privacy risk. Information regarding the variables is included through variable labels (VLs) in a DJ; VL is the name and/or meaning of the variables in the data. The variables and values in the data in a DJ are summarized as VLs. For example, the dataset on the "UNHCR Refugee Population Statistics" obtained from the Humanitarian Data Exchange (https://data.humdata.org/) includes the variables "country," "origin," "population type," "year," and "population," each of which contains values (Figure 1). Even if the data are not publicly available, we can learn and evaluate whether the data would be useful for our purposes using the data summary described in a DJ. Some data include private information, that is, values and variables such as "name," "age," or "address." The description framework of a DJ allows stakeholders to learn a summary of the data from the attributes mentioned in the DJ, thus, reducing the risks inherent to data management and privacy. The DJ has been introduced to support cross-disciplinary data exchange and collaboration in the creative workshop method, i.e., Innovators Marketplace on Data Jackets (IMDJ). For further details regarding the methodology and results of the IMDJ, see references [1,12,16].

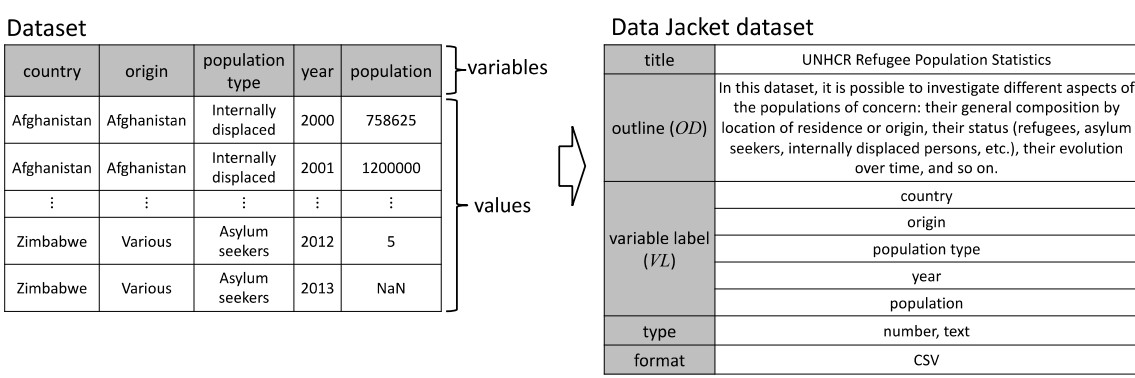

**Figure 1.** An example dataset and its corresponding data jacket (DJ).

VQ, however, focuses only on the variables in the data. The data possess other important attributes, namely, the types and formats. In the dataset on the "UNHCR Refugee Population Statistics," the data format is presented as a "CSV," whereas data types are "number" and "text," which are important attributes for stakeholders when considering data combinations. In this study, we extend the matrix-based method for the inference of variables to a method for inferring the data attributes. The motivation and the objective of our study are to infer the related attributes of the data (types, formats, and variables) from the data outlines presented as free text. We used a dataset of DJs as the training data. The significance of our approach and the contributions of our study can be summarized as follows: The proposed method is the first approach for inferring the data attributes while focusing

on the similarities of datasets using a data outline. Our method can infer the related data attributes from free text queries. In particular, it can be used not only for knowledge discovery from the data but also for decision-makers who wish to acquire new data. Our method can support the search for a useful set of variables, types, and formats used as data for decision making. Note that the definition of data in this study is a set of described abstracted events in the world. That is, as shown in Figure 1, the data consist of sets of variables with values. In contrast, the DJs in this study are the summary of the data consisting of attributes (types, formats, and variables) with elements.

The remainder of this paper is organized as follows: In Section 2, we briefly review the previous matrix-based method for inferring VLs and subsequently, formulate the proposed method and its inference procedures. In Section 3, we demonstrate the effectiveness of the proposed method by comparing its performance to other methods used for this purpose. Furthermore, we analyze the characteristics of the DJs and their attributes. In Section 4, we discuss the results obtained from the experiment. Finally, we provide some concluding remarks and discuss the areas of future work in Section 5. The notations used herein are summarized in Table 1.

**Table 1.** Notations.

| Symbol | Description |
| --- | --- |
| DJ | Summary of data in natural language (data jacket) |
| OD | Outline of data described in DJs (data outline) |
| VL | Name/meaning of variables in data (variable label) |
| $D$ | # DJs |
| $V$ | # VLs |
| $W$ | # terms in ODs |
| $L$ | Set of elements |
| $|L|$ | # elements |

## 2. Proposed Method

### 2.1. Models

The purpose of our method is to infer the possible attributes of data, namely, the types, formats, and variables, from free text queries. Let us define the meaning of the attributes and elements. The data attributes are the metadata characteristics of the data. In this study, we used variables, types, and formats as the attributes. The elements are the substances of the attributes. Table 2 shows some examples.

**Table 2.** Example elements in each attribute.

| Attribute | Example Elements |
| --- | --- |
| variable | latitude, longitude, address, weather, time, year |
| type | number, text, image, table |
| format | CSV, PDF, JSON, TXT, MOV |

The objective is to obtain sets of likely elements of attributes $\{l^{attr} \in L^{attr} | f_n(l^{attr}, OD_x)\}$ stored in the training data by inputting ODs ($OD_x$) as queries. Here, $f_n(l^{attr}, OD_x)$ represents a condition in which a set of the top $n$ elements ($l^{attr}$) are associated with $OD_x$. In addition, $L^{attr}$ is the set of elements and $attr \in \{type, format, variable\}$. We assume that the datasets are similar when the types of information used to explain the data (OD) are similar. In other words, when the similarity between ODs is high, the elements in each attribute are similar. For example, "the location data of elementary schools in Tokyo" and "the location data of the streetlights' installation in Tokyo" share the terms "location" and "Tokyo," and the data may share the same variables "latitude," "longitude," and "address." In addition, the variables may be of the same type, for example "latitude" and "longitude" are "numbers," and an "address" is a "text." The data might, therefore, be stored in the same format,

CSV. An important assumption in this study is that when the descriptive texts of a pair of datasets contain similar terms, they will have similar elements in each data attribute.

## 2.2. Preprocessing Steps

We used the bag-of-words and vector space model [17,18] to create a term-document matrix. In the preprocessing steps, we conducted a morphological analysis of the OD text by (1) extracting words, (2) removing stop words, and (3) restoring words to their original forms.

## 2.3. Similarity of Data from ODs (Term-Element Matrix)

We considered an algorithm that calculates the similarity between the training data of ODs. After conducting preprocessing for each OD, the ODs are converted into a term-OD matrix, $M = \{v_{ik}\}$ ($W \times D$), where $M$ comprises $D$-dimensional term vectors as rows, and $W$-dimensional OD vectors as columns. Each element $v_{ik}$ in an OD vector ($\boldsymbol{od}_k$) corresponds to the frequency at which a term (row $i$) occurs in an OD (column $k$). It is noteworthy that the subscript T at the upper-right corner of the vectors represents the transposition, and the matrices and vectors are highlighted in bold.

In the second step, a set of elements is converted into an element-OD matrix $R$. In the training data, ODs and elements in the attributes are linked when they appear in the same data. An element-OD matrix $R = \{r_{jk}\}$ ($|L| \times D$) consists of $|L|$-dimensional element vectors as rows and $D$-dimensional OD vectors as columns. Each element $r_{jk}$ in the $k$th OD vector corresponds to the frequency (0 or 1) at which the $j$th element is included in the $k$th OD.

In the third step, we create a term-element matrix $E = MR^{\mathrm{T}}$ ($W \times |L|$) from the term-OD matrix $M$ ($W \times D$) and element-OD matrix $R$ ($|L| \times D$) obtained in the second step. This process is equivalent to mapping the $i$th ($1 \leq i \leq |L|$) $D$-dimensional element vector in the OD space into a $W$-dimensional term space using the term-OD matrix $M$. The elements of the term-element matrix $E$ are represented as $e_{ij} = \sum_{k=1}^{D} v_{ik} r_{kj}$. In other words, the term-element matrix $E$ is equivalent to the adjacency matrix of a three-partite graph that consists of three disjoint sets of nodes, namely, terms, ODs, and elements (Figure 2). The element $e_{ij}$ of the term-element matrix $E$ represents the number of paths from the $i$th term ($t_i$) to the $j$th element ($l_j$) according to the OD nodes.

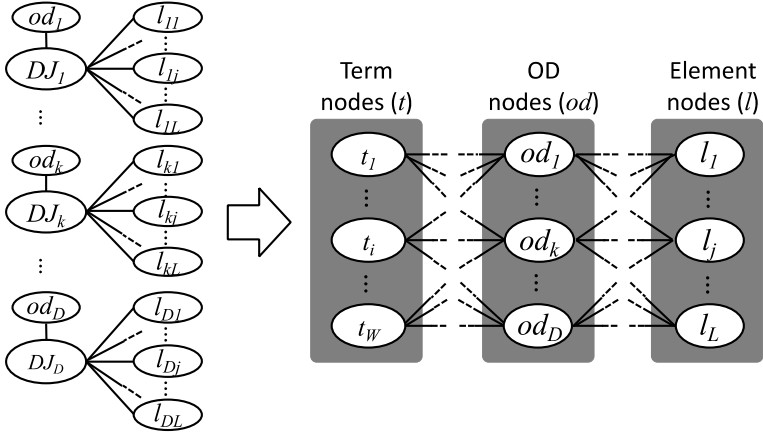

**Figure 2.** Adjacency matrix of the three-partite graph obtained by transforming DJs linked with data outlines (ODs) and elements.

The process above implements a function collectively to obtain a set of elements from free text queries as the term-element matrix $E$. Using this matrix, a scored set of the elements of the attributes is obtained by considering the similarity between ODs in the matrices $E$ and $OD_x$ whose elements are unknown. When $OD_x$ is given, a $W$-dimensional feature vector of $OD_x$ ($\boldsymbol{od}_x$) is obtained after the preprocessing of the morphological analysis. By comparing the similarity of $\boldsymbol{od}_x$ and each

$W$-dimensional feature vector of elements ($l_j$ ($1 \leq j \leq |L|$)) in matrix $E$, a scored set of elements is obtained.

## 2.4. Co-Occurrence of Elements (Term-Element Matrix)

We implemented an algorithm to calculate the co-occurrence of elements. A co-occurrence is a feature in which a highly frequent pair of elements appear simultaneously in the attributes. For example, "year" and "day," or "latitude" and "longitude," appear together frequently in the variable attributes. Moreover, in the type data attribute, "time series" and "number" appear together in the same data. Assuming that all pairs of elements in the same data occur once, we express the element co-occurrence matrix as $C = \{c_{ij}\}$ ($= RR^T$)($|L| \times |L|$). The element $c_{ij}$ represents the number of DJs that include a pair of elements $l_i$ and $l_j$, which is calculated as $c_{ij} = \sum_{k=1}^{D} r_{ik} r_{kj}$.

Finally, we obtain a term-element matrix $EC$ as a product of the term-element matrix $E$ and the element co-occurrence matrix $C$. The term-element matrix $EC$ consists of $V$-dimensional term vectors as rows, and $W$-dimensional element vectors as columns; this is the same structure as that of the term-element matrix $E$. The element $g_{ij}$ of matrix $EC$ is given as follows:

$$g_{ij} = \sum_{m=1}^{|L|} \left( \sum_{k=1}^{D} v_{ik} \, r_{km} \right) \left( \sum_{l=1}^{D} r_{ml} \, r_{lj} \right). \tag{1}$$

This represents a value that considers the similarities of the ODs and queries (the function of matrix $E$), and the co-occurrence of elements (the function of matrix $C$). The structure of the term-element matrix $EC$ is equivalent to the adjacency matrix of the five-partite graph. If two elements co-occur frequently, their weight in $EC$ will be increased by the matrix $C$, thereby increasing the final scores and ranks. When $OD_x$, whose elements of the attribute are unknown, is given, we obtain a $W$-dimensional feature vector of $OD_x$ ($\boldsymbol{od}_x$) from the corpus. By comparing the similarity of $\boldsymbol{od}_x$ and each $W$-dimensional feature vector of elements ($\boldsymbol{l}_j$) in the matrix $EC$, a scored set of elements is obtained.

## 3. Experimental Details

### 3.1. Purpose and Method

We evaluated the inference ability of elements in the attributes, namely, types, formats, and variables of data from free text queries by conducting an experiment. We introduced the string matching (TSM) and Doc2vec [19,20] as methods comparable to the proposed approach because a method based on string matching with elements of the attributes can be applied to a situation in which data are retrieved based on description. For example, when users search the related data from the query "the location data of the police stations in CSV," the string matching retrieves the variable "location" and the format "CSV." Therefore, the function of TSM ($f'_n(l^{attr}, OD_x)$)) can be used to obtain sets of elements ($\{l^{attr} \in L^{attr} | f'_n(l^{attr}, OD_x)\}$) stored in the training data that match the terms included in the ODs by inputting an OD ($OD_x$) as a query. The inputted ODs are converted into a bag-of-words similarly as in our proposed method. The elements obtained are scored in the descending order of the number of acquisitions.

Doc2vec is a paragraph vector model that learns a document representation by predicting the words that appear. In this experiment, the function of Doc2vec is to return a set of highly similar elements ($l^{attr}$) by comparing the feature vectors of each element ($\boldsymbol{l}^{attr}$) in the attributes by inputting an OD ($OD_x$) as a query. The parameters used for Doc2vec are a window size of eight, dimension of $d = 400$ and no downsampling. To create the learning model, we input each OD in the training data as a paragraph and obtain the feature vector of the ODs in 400 dimensions, which is different from matrices $E$ and $EC$. The models of matrices $E$ and $EC$ are based on one-hot encoding, namely, the dimensionality of the vectors is equivalent to the number of terms in the training data. The advantage

of using the word embedding method is reduced dimensionality. In the inferring process of Doc2vec, the query ($OD_x$) is converted into a 400-dimensional feature vector ($od_x$) by the learning model; feature vectors of elements ($l^{attr}$) of elements ($l^{attr}$) are obtained from the description of the elements through the learning model. We subsequently calculate the similarities of $od_x$ and $l^{attr}$, and obtain the sets of elements with similarity scores in the descending order.

The similarity scores of $od_x$ and $l_j^{attr}$ are calculated as cosine similarities given by $sim\left( od_x, \; l_j^{attr} \right) = od_x \cdot l_j^{attr} / |od_x| \left| l_j^{attr} \right|$, where $l_j^{attr}$ denotes the $j$th feature vector in the term-element matrix based on the attributes ($attr \in \{type, \; format, \; variable\}$). For weighing the discriminative terms in the DJs, we introduce the term frequency–inverse document frequency (tf-idf) in the weighting scheme [21], which identifies distinctive terms in each DJ. The term frequency (tf) represents the number of times a term appears in a document, and the inverse document frequency (idf) diminishes the weight of the frequent terms in all documents and increases the weight of terms that rarely appear.

We used leave-one-out cross-validation (LOOCV) for validation, and the precision, recall, and F-measure for evaluation. We define precision as $P = TP/(TP + FP)$ and recall as $R = TP/(TP + FN)$ using the top $n$ elements returned as the inferred results scored based on the similarities, where $TP$ = true positives, $FP$ = false positives, and $FN$ = false negatives. The F-measure is defined as $F = 2PR/(P + R)$. Finally, by calculating the average F-measure of each query, we compared the performances of our methods, string matching, and Doc2vec.

### 3.2. Attributes of Training Data (Corpus)

In this study, we used 1502 DJs collected from business persons, researchers, and data holders who are interested in using data for training in various domains (http://160.16.227.37/sparql). Twelve attributes were provided in the description of the DJs (e.g., title, outline, VLs, sharing policy, format) to identify the datasets [16]. We extracted the data outlines, and the types, formats, and variables of the data, from the DJ database using SPARQL queries (see Supplementary Materials). Because the description rule of the DJs did not mandate that information on all data attributes must be entered, a few DJs lacked some of the elements. Therefore, we could only use 1047 DJs including the attributes of the formats for the training data to infer the format elements, 1149 DJs including the type attributes to infer the elements of the types, and 1098 DJs including the variable attributes to infer the elements of the variables.

The statistics of the training (corpus) data are shown in Table 3. The corpus and dictionary were constructed using all words in the ODs. We removed punctuation marks and symbols in the texts, restored words to their original forms, and extracted nouns, verbs, adverbs, and adjectives. For a morphological analysis, we used MeCab [22], a typical tool for analyzing morphemes of Japanese texts, and the Natural Language Toolkit (NLTK) (https://www.nltk.org/) for English texts. We removed auxiliary verbs, symbols, and extremely frequent nouns, such as "data" and "information." The title descriptions of the DJs occasionally exhibited insufficient information on the data. We combined the description of the titles and outlines of the DJs when the retrieval system searched the related data using queries.

**Table 3.** Statistics of the training data.

| | |
|---|---|
| Number of DJs | 1502 |
| Total number of terms in DJs | 38,722 |
| Unique terms in DJs | 7886 |
| Total number of formats in DJs | 1421 |
| Unique formats in DJs | 48 |
| Total number of types in DJs | 2956 |
| Unique types in DJs | 18 |
| Total number of VLs in DJs | 7552 |
| Unique VLs in DJs | 5559 |

The format attribute comprised 48 elements. Figure 3 shows the number of DJs based on format. Note that some data are provided in multiple formats; for example, official data are available in PDF, CSV, or TXT files. Therefore, the numbers in the figures contain duplicates. Because the data have 1.36 formats (maximum of 8, minimum of 1) on average, to calculate the precision, recall, and F-measure, we used the top-five similar formats in this experiment. In contrast, the type attribute comprises 18 elements. Figure 4 shows the number of DJs based on type. Because the data have several types, the numbers in the figures also contain duplicates. Because the data have 2.57 formats (maximum of 8, minimum of 1) on an average, to calculate the precision, recall, and F-measure, we used the top-five similar types.

However, there were approximately 5600 variables in the training data. Figure 5 shows the number for the top-15 VLs in the training data. The distribution of variables consisted of a few extremely frequent variables and many variables with lower frequencies, which conforms to the power law distribution (Figure 6). Therefore, we compared the performances of our method with that of other methods based on the frequency of the VLs.

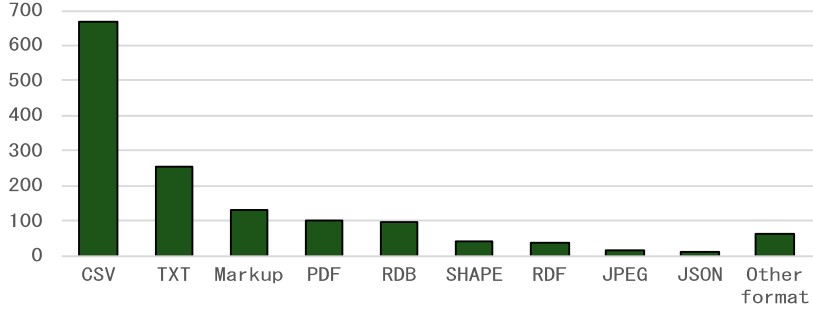

**Figure 3.** Number of DJs by format.

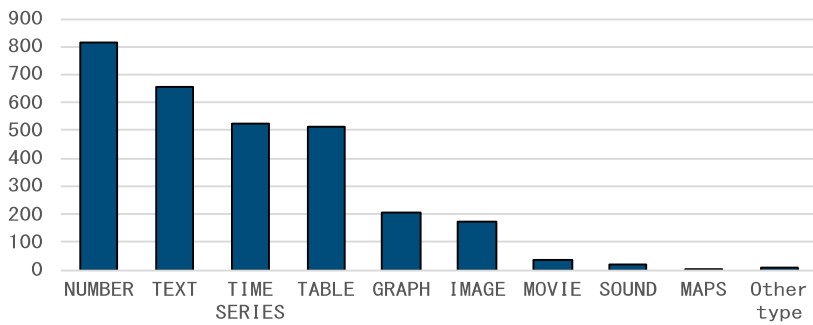

**Figure 4.** Number of DJs by type.

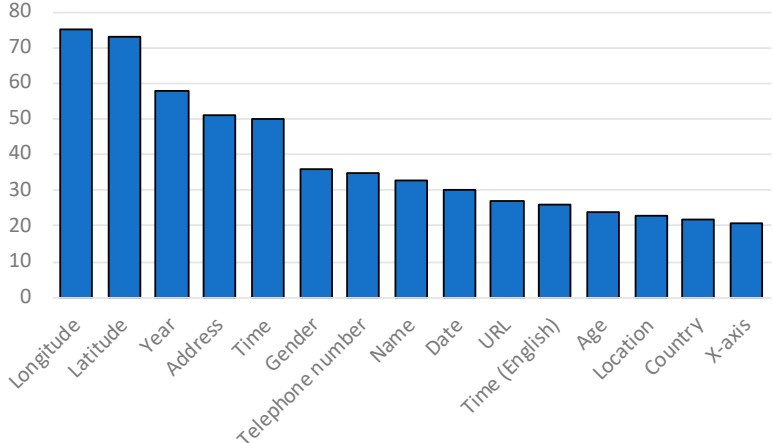

**Figure 5.** Top-15 variable labels (VLs) in the corpus data.

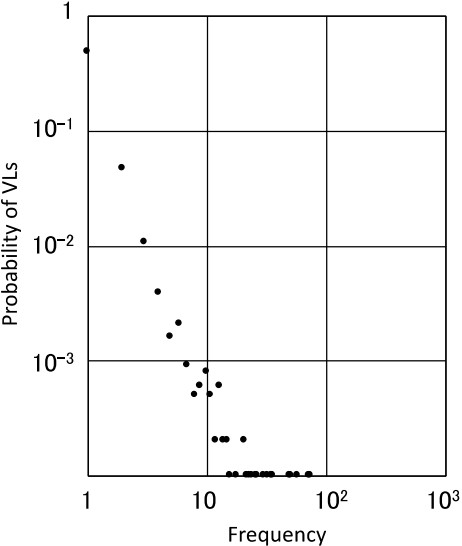

**Figure 6.** Distribution of VLs.

## 4. Result and Discussion

Tables 4–6 show the evaluation results for each data attribute. With respect to the results related to the formats and types, we listed the top-five elements returned as the inferred results scored based on the similarities from each query. Furthermore, we present the top-ten VLs returned as the inferred results scored based on similarities from each query using our proposed matrices *E* and *EC*, as well as TSM and Doc2vec.

By comparing the F-measures calculated from the precision and recall of each method, we observed that the results inferred using the matrices *E* and *EC* demonstrated a better performance in inferring the type, format, and variable elements. In particular, the performance of matrix *E* was the best in terms of the F-measure score. The results indicate that, although the data outline is an important attribute for characterizing the data, it does not always include information regarding the attributes. In other words, the string matching of ODs and each element in the attributes is insufficient to infer the elements. The performance of Doc2vec is comparatively poor. A reason is that the ODs contain few terms that describe the type, format, and variable elements. We compared the commonality of terms derived from the ODs in the corpus of the training data with the VLs, formats, and types. Subsequently, only 162 out of 7871 terms were in common with the VLs. That is, only 162 words in the ODs contributed to the discovery of the VLs. If the commonality of the terms is low, we cannot sufficiently compute the similarity even if the dimensionality of the word embedding is low compared to the one-hot vectors. In contrast, the formats and types were both included in the ODs. Consequently, the F-measures of the formats and types using TSM and Doc2vec are higher than those of the VLs.

According to Hayashi and Ohsawa [15], the performances of the term-element matrices *E* and *EC* are almost the same. However, when comparing the F-measures of the results, we found significant differences between the matrices *E* and *EC* for each attribute. Using a paired t-test, we obtained $t(2092) = 5.0$, $p < 0.01$ for the format results; $t(2296) = 15.9$, $p < 0.01$ for the type results; $t(2194) = 7.28$, $p < 0.01$ for the variable results. From this experiment, we concluded that a model based on the idea that "a pair of datasets whose similarity of outlines is high are similar in terms of having similar elements in the attributes," namely, the effect of matrix *E*, is suitable for inferring elements in the data attributes. In other words, the information on other datasets (the relationship between the ODs and elements in each attribute) may compensate well for the missing terms in explaining the data and may be suitable for discovering elements from the outlines of data whose elements are missing in the attributes.

The inferred examples using the co-occurrence model are exemplified in the study by Hayashi and Ohsawa; the results are not contrary to human intuition. However, when evaluating the performance mechanically, the model considering only the similarity of the ODs, which does not consider the co-occurrence of elements, is better. It is thought that the frequency distribution of the elements is not a Gaussian distribution but a power distribution. Although a few types of formats and types exist, "CSV," "TXT," "number," and "text" are relatively large compared to the other elements (Figures 3 and 4). Moreover, more types of VLs exist, and it is clear that the distribution influences performance. Therefore, in this study, we conducted a detailed analysis with a threshold value for the VLs.

**Table 4.** Results of formats (average scores ± standard deviation).

|  | F-Measure | Precision | Recall |
|---|---|---|---|
| Matrix $E$ | **0.266 ± 0.190** | **0.664 ± 0.443** | **0.174 ± 0.141** |
| Matrix $EC$ | 0.250 ± 0.203 | 0.619 ± 0.463 | 0.164 ± 0.152 |
| TSM | 0.046 ± 0.162 | 0.034 ± 0.124 | 0.078 ± 0.281 |
| Doc2vec | 0.003 ± 0.104 | 0.089 ± 0.261 | 0.026 ± 0.071 |

**Table 5.** Results of types (average scores ± standard deviation).

|  | F-Measure | Precision | Recall |
|---|---|---|---|
| Matrix $E$ | **0.530 ± 0.247** | **0.816 ± 0.304** | **0.423 ± 0.243** |
| Matrix $EC$ | 0.463 ± 0.269 | 0.712 ± 0.364 | 0.371 ± 0.251 |
| TSM | 0.089 ± 0.203 | 0.079 ± 0.178 | 0.120 ± 0.313 |
| Doc2vec | 0.179 ± 0.182 | 0.276 ± 0.309 | 0.144 ± 0.151 |

**Table 6.** Results of variables (5559 types of variable labels (VLs)) (Average scores ± standard deviation).

|  | F-Measure | Precision | Recall |
|---|---|---|---|
| Matrix $E$ | **0.110 ± 0.210** | **0.165 ± 0.311** | **0.095 ± 0.191** |
| Matrix $EC$ | 0.089 ± 0.180 | 0.131 ± 0.264 | 0.078 ± 0.169 |
| TSM | 0.041 ± 0.096 | 0.034 ± 0.078 | 0.060 ± 0.145 |
| Doc2vec | 0.001 ± 0.008 | 0.001 ± 0.013 | 0.001 ± 0.009 |

As shown in Table 3, approximately 5600 types of variables exist in the training data, and the number of dimensions becomes extremely large when we create the term-VL matrices $E$ and $EC$. As discussed in the previous section, the distribution of variables consists of a few extremely frequent variables and many variables with low frequencies. Therefore, we compare the performance based on the threshold of the variable frequencies. Figures 7–11 show the boxplots of the F-measure using VLs appearing once, more than once, twice, thrice, and four times. The dots represent the mean values, and the lines inside each box represent the median. The top of the box is the first quartile, and the bottom is the third quartile. The bar on the top is the maximum value, and the bar on the bottom is the minimum value.

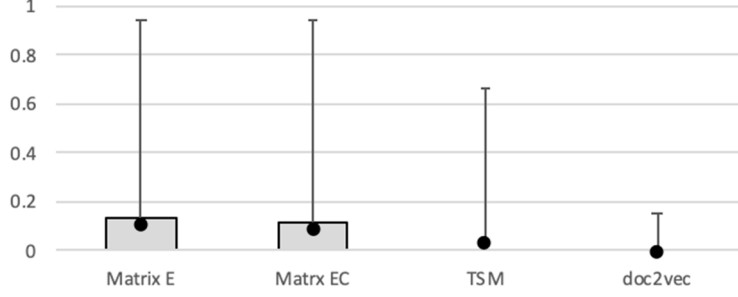

**Figure 7.** Boxplots of F-measure using all VLs (5559 types of VLs).

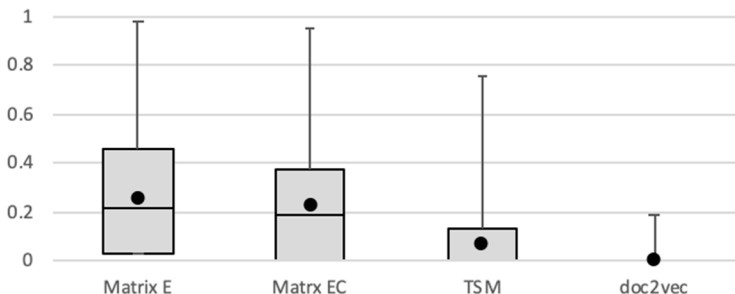

**Figure 8.** Boxplots of F-measure using variables appearing more than once (712 types of VLs).

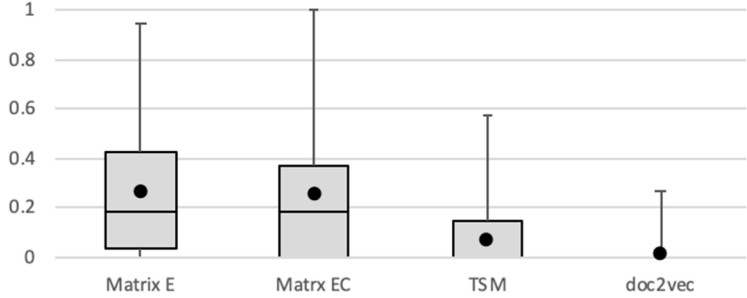

**Figure 9.** Boxplots of F-measure using variables appearing more than twice (245 types of VLs).

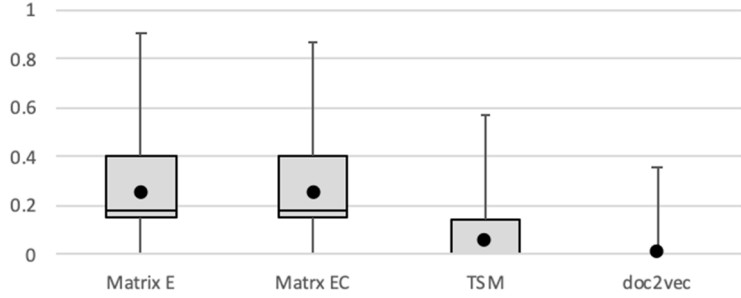

**Figure 10.** Boxplots of F-measure using variables appearing more than thrice (139 types of VLs).

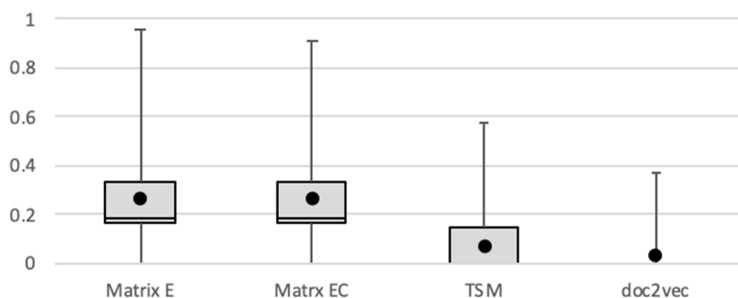

**Figure 11.** Boxplots of F-measure using variables appearing more than four times (100 types of VLs).

The types of VLs decrease according to the power law shown in Figure 6. For all VLs (Figure 7), the maximum F-measure is 0.947 for both matrices *E* and *EC*. Because the medians are zero, small numbers of highly frequent VLs affect the performance, and the F-measures of most of the data are low. This is because the types of VLs are diverse, and the low-frequency VLs occupy the majority. Hence, the means and medians of the F-measures increase for all methods by setting the threshold and adjusting the frequency VLs, as shown in Figures 8–11.

The results indicate that all performances improved by reducing the number of unique VLs; the F-measures generally improved until the threshold reached two (Figures 8–10). When the threshold

is three, however, almost no difference between matrices $E$ and $EC$ ($t(1282) = 1.50$, $p = 0.070$) are shown, and the result of matrix $EC$ increases to a higher level than that of matrix $E$. The results indicate that the method considering the co-occurrence of VLs may be suitable when using VLs that appear frequently with each other. A larger number of variables indicates that more noise may be included in the training data, which affects the inference performance.

In contrast, when we set the threshold to four, the F-measures of all methods tend to decrease. Reducing the number of variables represents a reduced amount of test data. In other words, although the performance for some of the data improves, it may be difficult to infer the variables of data that contain less frequent variables. These results suggest that using only 100 types of VLs is insufficient for inferring the VLs when the threshold is five.

## 5. Conclusions

We herein proposed a matrix-based method for inferring the elements in the attributes of data from the OD whose attribute elements (types, formats, and variables) are missing or unknown. We extended a previously proposed method by adding other attributes to identify the data in the description of the DJs, namely, the types, formats, and variables. When information is retrieved from data, string matching using the elements in the data attributes can be considered. However, free text queries do not always include terms corresponding to the data elements. Decision-makers who wish to acquire new data cannot discover information regarding what types of data should be obtained for their decision making. The proposed method will be helpful for encouraging data acquisition and for knowledge discovery.

With the proposed method, natural language processing using bag-of-words and a vector space model was used to calculate the similarity of the ODs. However, natural language processing was not used for estimating the similarity of elements. The ODs were small but include a certain number of terms; therefore, the similarities in the vector space model could be discussed and compared by creating the term-document matrix. However, the type, format, and variable elements were small and composed of one or several words. In future study, we aim to construct a model that considers the meaning of the elements and synonyms, even if they contain brief descriptions.

**Supplementary Materials:** Data from Data Jacket with Variable Labels and the results used to support the findings of this study were deposited in the Data Jacket Store repository in RDF/XML (http://160.16.227.37/sparql).

**Author Contributions:** Conceptualization of VARIABLE QUEST, T.H. and Y.O.; software, T.H.; formal analysis, T.H.

**Funding:** This research was funded by JST-CREST Grant Number JPMJCR1304, and JSPS KAKENHI Grant Numbers JP16H01836 and JP16K12428.

**Acknowledgments:** We wish to thank Editage (www.editage.jp) for providing English language editing.

**Conflicts of Interest:** The authors declare no conflicts of interest.

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
