# Peer review of "Matrix-Based Method for Inferring Elements in Data Attributes Using a Vector Space Model"

_information, doi:10.3390/info10030107_

Reviewer 1 Report
This paper tries to infer the related attributes of the data (types, formats, and variables) from the data outlines given as free text using a dataset of DJs as the training data. 

I tend to accept this paper. Some suggestions are as below:

1. Data is a big concept. It is better to give a clear definition in this paper.

2. It is a little difficult to read this paper so it is better to get help from a local english speaker.

3. It is better to introduce more background about DJs, what the relation is between this paper and DJs, and to clearly claim what the motivations, problems, challenges are. Or I think the readers who do not have relevance knowledge on DJs are hard to understand this paper.

Author Response

Title of the paper:

Matrix-based Method for Inferring Elements in the Attributes of Data Using a Vector Space Model

 Dear Reviewers,

I really appreciate editor’s support and the kind review from the reviewers. According to the reviewers’ comments, we modified the following points. Thank you again for your insightful and valuable comments.

 In response to Reviewer 1:

1.     Comment 1:

Data is a big concept. It is better to give a clear definition of this paper.

Answer:

We added the definition of data in this paper as “a set of described abstracted events in the world” and made the difference between data and the data jackets clearer (line 86-89).

2.     Comment 2:

It is a little difficult to read this paper so it is better to get help from a local English speaker.

 Answer:

According to the reviewer’s comment, we used the Editage (www.editage.jp) for providing English language editing.

3.     Comment 3:

It is better to introduce more background about DJs, what the relation is between this paper and DJs, and to clearly claim what the motivations, problems, challenges are. Or I think the readers who do not have relevant knowledge on DJs are hard to understand this paper.

 Answer:

Thanks to the reviewer’s comment, we added the explanation and the role of DJs (line 70-73 and 86-89), and showed the references for further understanding. The motivation and the problems we wanted to solve are shown in line 79-86.

END

Reviewer 2 Report

The paper presents a straightforward application of word embedding into data jackets.

Specifically, the embedding matrix E (term x elements) is computed from the product of (term x doc) and (doc x element) training set.

A new/test doc (Data outline) is compared to the (term x element) embedding matrix to find a ranked list of similar elements, the top K of which represent the likely elements contained in the term-doc.

The authors further enhanced the embedding matrix by considering co-occurence of elements, by multiplying the embeding matrix E by C, where C denotes the element x element correlation matrix. The enhanced embedding is known as EC.

From the results, EC does not perform as well as E.

E performed better than Doc2vec, because the embedding used for matrix E is the full 5559 elements/VLs, versus doc2vec's 400 reduced dimensions.

If you increase doc2vec's embedding dimension to 1000, 2000 or even 5000, the doc2vec results should improve.

Suggestions:

        W: # word terms in OD

        V: # variable labels / elements

        D: # data jackets   

pp.4 line 117-120

    to make for easier reading, suggest using unique subscripts i, j, k

    e.g.    M = {v_{ik}}        (W x D)

            R = {r_{jk}}        (V x D)

    this transition more directly and smoothly to e_ij in pp.4 line 131

            term-VL matrix

            E = MR^T            (W x V)

pp.4 line 134 - 135, figure 2

    likewise, in figure 2, suggest you use consistent/standardized subscripts

    e.g.,

        Term nodes (t_i)    (correct, no change here)

        OD nodes (od_k)     (correct, no change here)

        element nodes (l_j) (change from m to j subscript)

                      (l_V) (change from L to V)

Here in figure 2,  the number of elements is suddenly reduced from V to L, please explain why, e.g., selecting the top-L elements from V

pp.5

    figure 3 is not really helpful in explaining the role of C in EC, since the 5-partite graph only makes it harder to understand. Instead you can say that if two elements co-occurs frequently, their weight in EC will be boosted by C, thereby increasing their (element) final scores and rank.

Author Response

Title of the paper:

Matrix-based Method for Inferring Elements in the Attributes of Data Using a Vector Space Model

Dear Reviewers,

I really appreciate editor’s support and the kind review from the reviewers. According to the reviewers’ comments, we modified the following points. Thank you again for your insightful and valuable comments.

In response to Reviewer 2:

1.     Comment 1:

Suggestions:

W: # word terms in OD

V: # variable labels / elements

D: # data jackets

Answer:

According to the reviewer’s comment, we modified the description in Table 1.

2.     Comment 2:

to make for easier reading, suggest using unique subscripts i, j, k

    e.g.    M = {v_{ik}}        (W x D)

            R = {r_{jk}}        (V x D)

this transition more directly and smoothly to e_ij in pp.4 line 131

term-VL matrix

            E = MR^T            (W x V)

likewise, in figure 2, suggest you use consistent/standardized subscripts

    e.g.,

        Term nodes (t_i)    (correct, no change here)

        OD nodes (od_k)     (correct, no change here)

        element nodes (l_j) (change from m to j subscript)

                      (l_V) (change from L to V)

Here in figure 2,  the number of elements is suddenly reduced from V to L, please explain why, e.g., selecting the top-L elements from V

Answer:

We modified the descriptions of the matrices M, R, and E, according to the reviewer’s advice. Also, we modified Figure 2. We the range of the element l was the typo. The correct range is 1<=j<=|L|, we did not change l_L. We modified the range and the size of the matrices (line 138, 141, 142, 143, and 154).

3.     Comment 3:

figure 3 is not really helpful in explaining the role of C in EC, since the 5-partite graph only makes it harder to understand. Instead, you can say that if two elements co-occur frequently, their weight in EC will be boosted by C, thereby increasing their (element) final scores and rank.

 Answer:

According to the comment, we deleted Figure 3 and the sentence to explain the figure and added a detailed explanation about the function of the matrix C (line 175-177).

END